# Reply to Birch et al. Comment on “Takakura et al. Acupuncture for Japanese Katakori (Chronic Neck Pain): A Randomized Placebo-Controlled Double-Blind Study. *Medicina* 2023, *59*, 2141”

**DOI:** 10.3390/medicina60091438

**Published:** 2024-09-03

**Authors:** Nobuari Takakura, Miho Takayama, Ted J. Kaptchuk, Jian Kong, Hiroyoshi Yajima

**Affiliations:** 1Department of Acupuncture and Moxibustion, Tokyo Ariake University of Medical and Health Sciences, 2-9-1 Ariake, Koto-ku, Tokyo 135-0063, Japan; takayama@tau.ac.jp (M.T.); yajima@tau.ac.jp (H.Y.); 2Japan School of Acupuncture, Moxibustion, and Physiotherapy, 20-1 Sakuragaokacho, Shibuya-ku, Tokyo 150-0021, Japan; 3Program in Placebo Studies & Therapeutic Encounter, Beth Israel Deaconess Medical Center, Harvard Medical School, 330 Brookline Avenue, Boston, MA 02215, USA; ted_kaptchuk@hms.harvard.edu; 4Department of Psychiatry, Massachusetts General Hospital, Harvard Medical School, Charlestown, MA 02129, USA; kongj@nmr.mgh.harvard.edu

We are writing in response to the comment [1] made on “Acupuncture for Japanese Katakori (Chronic Neck Pain): A Randomized Placebo-Controlled Double-Blind Study”.

In this study, we investigated the effects of acupuncture on neck stiffness using two forms of double-blinded placebo needles, including a double-blind skin-touch placebo needle and a double-blind no-skin-touch placebo needle [2]. Please note that the skin-touch placebo needle (i.e., Streitberger placebo needle or Park sham acupuncture needle) has been widely used in acupuncture research [3,4,5,6,7,8,9,10,11,12]. This design allowed us to elucidate the specific effects of penetrating the skin and muscles (skin-piercing acupuncture versus skin-touching acupuncture) and the specific effects of skin-touch (skin-touch versus no-skin-touch placebo needle). We found that there was no significant difference between the penetrating and skin-touch treatments, as well as between the skin-touch and no-touch treatments for stiffness. Also, the change in stiffness due to the no-touch treatment was significantly larger than that of the no-treatment control. These results should address the claim at the end of the second paragraph in a comment [1].

Thus, we believe that our conclusions are well-supported by the data and results. The knowledge gained from this study could significantly enhance our understanding of acupuncture, although our findings are specific to the short-term effects on shoulder stiffness and should not be generalized to the efficacy of acupuncture in treating other conditions or disorders. Importantly, we have provided sufficient details about the study, enabling readers to form their own informed opinions without being misled.

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
