# Peer review of "Reply to Birch et al. Comment on “Takakura et al. Acupuncture for Japanese Katakori (Chronic Neck Pain): A Randomized Placebo-Controlled Double-Blind Study. Medicina 2023, 59, 2141”"

_medicina, 2024, doi:10.3390/medicina60091438_

Round 1

Reviewer 1 Report

Comments and Suggestions for Authors

Below are general comments and impressions regarding the “Response to the comment” letter to the editor.  

This lengthy response to a reader comment is mostly a repetition of the original study and only addresses the reader’s comments in a round-about and limited manner. Except for lines 68-74, which are clearly written, the rest of the manuscript is wordy, grammatically problematic, repetitive, and often tedious to follow. Subsequently, the article reads as if it is of first draft quality. It takes much effort to tease out the argument the author is trying to convey.

The list of citations (lines 9-12) is disruptive, it would be appreciated if they were listed as citations at end of letter.

Quotations are frequently used (lines 4, 44-45, 46, 64-65) but are not clear where they came from.

Overall, it seems that the authors are patting themselves on the back for a job-well-done and not taking the reader’s comments seriously. Subsequently, it is unlikely that this response will satisfy the commenter and will most likely exacerbate division in the field.

Comments on the Quality of English Language

Quality of English can be much improved. Detailed in main comments attachment. 

Author Response

Thank you for your comments on the improvement of our manuscript.

In accordance with your comment, we have revised the manuscript. Regarding the references in the text, we have left them as they are, as we would like to include them in the main text. I hope that you understand this.

Comment: 

This lengthy response to a reader comment is mostly a repetition of the original study and only addresses the reader’s comments in a round-about and limited manner. Except for lines 68-74, which are clearly written, the rest of the manuscript is wordy, grammatically problematic, repetitive, and often tedious to follow. Subsequently, the article reads as if it is of first draft quality. It takes much effort to tease out the argument the author is trying to convey.

The list of citations (lines 9-12) is disruptive, it would be appreciated if they were listed as citations at end of letter.

Quotations are frequently used (lines 4, 44-45, 46, 64-65) but are not clear where they came from.

Overall, it seems that the authors are patting themselves on the back for a job-well-done and not taking the reader’s comments seriously. Subsequently, it is unlikely that this response will satisfy the commenter and will most likely exacerbate division in the field.

Response:

We revised our manuscript according to the comments except for the list of citations (lines 9-12).